# Detection and Mitigation of Neurovascular Uncoupling in Brain Gliomas

**DOI:** 10.3390/cancers15184473

**Published:** 2023-09-08

**Authors:** Shruti Agarwal, Kirk M. Welker, David F. Black, Jason T. Little, David R. DeLone, Steven A. Messina, Theodore J. Passe, Chetan Bettegowda, Jay J. Pillai

**Affiliations:** 1Division of Neuroradiology, Russell H. Morgan Department of Radiology and Radiological Science, Johns Hopkins University School of Medicine, Baltimore, MD 21287, USA; sagarwal.jhmi@gmail.com; 2Division of Neuroradiology, Department of Radiology, Mayo Clinic Rochester & Mayo Clinic College of Medicine and Science, Rochester, MN 55905, USA; welker.kirk@mayo.edu (K.M.W.); black.david@mayo.edu (D.F.B.); little.jason@mayo.edu (J.T.L.); delone.david@mayo.edu (D.R.D.); messina.steven@mayo.edu (S.A.M.); passe.theodore@mayo.edu (T.J.P.); 3Department of Neurosurgery, Johns Hopkins University School of Medicine, Baltimore, MD 21287, USA; cbetteg1@jhmi.edu

**Keywords:** BOLD fMRI, presurgical mapping, neurovascular uncoupling, cerebrovascular reactivity, brain tumors, gliomas

## Abstract

**Simple Summary:**

We discussed the reliability of fMRI for in vivo evaluation of eloquent cortex, which helps to reduce the risk of postsurgical morbidity. Task-based fMRI (tb-fMRI) is commonly utilized for the noninvasive, accurate evaluation of eloquent cortex implicated in sensorimotor, language, and visual function. Resting-state fMRI (rs-fMRI) is being increasingly employed in addition to tb-fMRI and may be able to serve as an adjunct for tb-fMRI for presurgical mapping in the future. There is, however, a need to confirm the reliability of rs-fMRI data in various clinical contexts, and more validation and standardization is necessary prior to adopting this tool for daily clinical care. Although tb-fMRI has been proved to be a trustworthy method, the problem of neurovascular uncoupling (NVU) needs to be addressed to give the most accurate functional evaluation in clinical practice. Breath-hold cerebrovascular reactivity (BH-CVR) methods can be used to assess risk of NVU. We hope that future research will investigate and overcome the problem of NVU encountered in presurgical brain mapping with BOLD fMRI.

**Abstract:**

Functional magnetic resonance imaging (fMRI) with blood oxygen level-dependent (BOLD) technique is useful for preoperative mapping of brain functional networks in tumor patients, providing reliable in vivo detection of eloquent cortex to help reduce the risk of postsurgical morbidity. BOLD task-based fMRI (tb-fMRI) is the most often used noninvasive method that can reliably map cortical networks, including those associated with sensorimotor, language, and visual functions. BOLD resting-state fMRI (rs-fMRI) is emerging as a promising ancillary tool for visualization of diverse functional networks. Although fMRI is a powerful tool that can be used as an adjunct for brain tumor surgery planning, it has some constraints that should be taken into consideration for proper clinical interpretation. BOLD fMRI interpretation may be limited by neurovascular uncoupling (NVU) induced by brain tumors. Cerebrovascular reactivity (CVR) mapping obtained using breath-hold methods is an effective method for evaluating NVU potential.

## 1. Basics of BOLD fMRI

Blood oxygen level-dependent (BOLD) functional magnetic resonance imaging (fMRI) was originally described by Ogawa et al., in 1992 [1] using the magnetic characteristics of deoxygenated (strongly paramagnetic) and oxygenated (weakly diamagnetic) blood. When a neuron is activated, the cell quickly absorbs additional oxygen from nearby capillaries, resulting in a larger proportion of deoxyhemoglobin in the downstream venule. Based on these regional increases in paramagnetic deoxyhemoglobin, we would expect a large decrease in MR signal in the active brain area. The presence of neurotransmitters, on the other hand, causes the surrounding support cells, astrocytes, to release vasoactive substances into the arterioles, allowing their dilatation to effectively provide more oxygen and nutrient-rich blood to the functioning regional neuron. This mismatch between increased neuronal oxygen consumption and greater compensatory vasodilatation, with resultant greater increase in regional blood flow, is the basis of the BOLD principle. Given the weakly diamagnetic nature of oxyhemoglobin, the local magnetic resonance signal is higher in the area of neuronal activity when compared to inactive neurons wherein adjacent venules contain mostly strongly paramagnetic deoxyhemoglobin. The relative concentration of oxyhemoglobin in the downstream venules determines the signal resolution between active and resting neurons. Therefore, this approach is known as blood oxygen level dependent functional MRI.

## 2. BOLD fMRI Acquisition

At 3T or lower field strengths, which are typically used for clinical fMRI, the T2* shortening effect is the major mechanism for BOLD contrast [2,3]. T2*-weighted gradient echo (GRE) sequences [4], as well as single-shot imaging sequences such as echo planar imaging (EPI) [5], are utilized to acquire BOLD fMRI at magnetic fields of 3T or below. “Single-shot” sequences, which employ rapid gradient switching technologies to enable the recording of data for a full slice in one readout window following one excitation, accomplish the high sampling rates required for fMRI. The multiband parallel imaging [6] approach is one of the most recent developments. Multiband EPI entails excitation of several 2D slices at the same time, followed by reconstruction of the constituent slices.

## 3. Preoperative BOLD fMRI

Patients with brain tumors, vascular malformations such as arteriovenous malformations or cavernous malformations, or focal epileptogenic brain lesions such as malformations of cortical development or gray matter heterotopias, who are undergoing presurgical planning, commonly undergo clinical fMRI exams [7,8]. Preoperative fMRI has proven useful in guiding neurosurgeons in decisions regarding safety of complete or partial surgical resection of the lesion, as well as in selecting patients for awake vs. asleep intraoperative mapping during craniotomy. Preoperative identification of potential functional cortical regions surrounding a lesion may help guide planning for, as well as efficient performance of, intraoperative cortical stimulation mapping. This results in reduced overall surgical time and allows for the safest surgical options.

Although not all neurosurgeons rely on fMRI, most consider fMRI to be the current standard of care for presurgical planning. This widespread adoption of clinical fMRI has been further facilitated by the development of Current Procedural Terminology (CPT) codes in the United States that allow for clinical billing for these types of examination.

Not only is fMRI helpful for localization of eloquent cortex, particularly in the language, sensorimotor, and visual systems, but it is also useful for language lateralization, i.e., hemispheric language dominance determination.

## 4. Clinical Evaluation of Brain Gliomas with Task-Based BOLD fMRI

During a presurgical clinical BOLD fMRI evaluation, the patient is instructed to execute various activities to elicit neuronal activation within particular brain networks (such as finger tapping for motor activation or silent word generation for language activation) while in the scanner. Task-based fMRI (tb-fMRI) is the name given to this kind of fMRI examination [9]. Repeated scanning of the brain is performed during a tb-fMRI scan session when the person performs a control task or an actual intended task involving motor, language, or visual function that involves application of repeated stimuli. Block designs, which generally span between 2 and 4 min with periodic epochs of stimulation alternating with epochs of control conditions, referred to as “on” (active) and “off” (control) conditions generally lasting from 15 to 30 s, are the most widely utilized paradigms for clinical presurgical brain mapping. The activation of interest is due to the difference between the BOLD signal detected during the “on” phase and the signal obtained during the “off” phase. Presurgical mapping with tb-fMRI has already been established as standard of care at many academic centers around the world.

## 5. Resting-State Functional Connectivity and Neuronal Activity

Resting-state BOLD fMRI (rs-fMRI), which does not require patient performance of any specific tasks like finger tapping for motor function [10], is becoming increasingly popular for both research and selected clinical applications. In contrast to tb-fMRI, in rs-fMRI the patient just lies in the scanner without performing any particular task, and the spontaneous BOLD signal variations that represent brain activity are recorded and analyzed. In rs-fMRI, various regions of the brain exhibit synchronized low frequency fluctuations in BOLD signals over time, independent of any specific task, allowing for the extraction of intrinsic brain networks referred to as resting-state networks [10,11,12]. Resting-state functional connectivity network mapping is a promising method for studying the functional structure of the healthy and diseased brain, especially in people who are unable to perform difficult behavioral tasks. The field of presurgical brain mapping has seen significant progress in the use of rs-fMRI [13,14,15,16,17,18,19,20].

Rs-fMRI has not yet been widely accepted for the purpose of presurgical mapping due to high inter-subject variability in identification of networks as well as absence of standardization of analysis methods [21,22]. Additional problems include absence of multi-institutional validation of analysis methods for rs-fMRI clinical applications, as well as operator dependence of interpretation of even data-driven analysis methods such as independent component analysis (ICA) [23]. The commonly employed methods to identify resting-state networks are seed-based correlation analysis and ICA. Seed-based analysis involves correlating the average BOLD time course of voxels within a seed region with that of all other voxels throughout the brain. On the other hand, independent component analysis evaluates the statistical independence between all of the voxels’ BOLD time courses by dividing them into several spatial components [23]. However, there is no obvious connection between the components discovered and the various brain activities [24]. Cluster analysis and graph theoretic analysis are further functional connectivity analysis techniques for rs-fMRI investigation which primarily examine the similarities in time series of low frequency fluctuations among different brain regions, but these methods have not been as useful clinically given the relatively abstract and less anatomically-based nature of analysis, particularly with respect to graphs.

Studies have shown that the relative strength of rs-fMRI low-frequency fluctuations can vary across brain regions, making it a potential indicator of individual differences or dysfunction. Another rs-fMRI metric, the amplitude of low-frequency fluctuations (ALFF), captures the total power within the frequency range of 0.01 to 0.08 Hz, and therefore, serves as an index for the strength of low-frequency BOLD signals. Interestingly, contrary to the functional connectivity metrics, ALFF is commonly regarded as a neuronal activity metric [25]. The regional homogeneity (ReHo) metric is another voxel-based measure of brain activity in rs-fMRI which assesses the synchronization between the low frequency signals of a particular voxel and its neighboring voxels [26]. These methods (ALFF and ReHo) allow whole-brain voxel-wise analysis of rs-fMRI signals without interrogation of specific networks.

## 6. Neurovascular Uncoupling (NVU) and Its Impact on Clinical fMRI

BOLD signal measures hemodynamic changes that are indirect measures of neuronal activity [27]. In the vicinity of tumors or vascular malformations, the normally tight coupling [28] between neuronal activity and resultant hemodynamic changes occurring in the adjacent vasculature get disrupted. The reduced capacity for dynamic blood flow control in response to increasing neural demand near tumors can serve as a biomarker for potential neurovascular uncoupling (NVU) [29].

The neurovascular coupling sequence encompasses a series of biochemical processes at the cellular level responsible for autoregulation of blood flow [28,30]. Tumors may alter the extracellular matrix, endothelial cells, as well as pericytes and astrocytic end feet connected to arterioles, which would interfere with the normal neurovascular coupling sequence and adversely affect the ability of such arterioles to dilate or constrict in response to vasoactive stimuli or adjacent neuronal activation [31,32,33,34]. NVU can occur when the normal neurovascular coupling sequence is disrupted. Tumor angiogenesis is believed to be the primary cause of the NVU surrounding high-grade gliomas [35,36]. Gliovascular uncoupling involving the rupture of astrocytic connections with the surrounding microvasculature [37,38,39], may contribute to the development of NVU in cases of low-grade gliomas.

NVU can limit the usefulness of tb-fMRI activation measurements in patients with tumors, as NVU can cause decreased or absent activations on tb-fMRI mapping, even when the patient can adequately perform the required tasks that typically activate the relevant networks of interest [40,41]. The possibility of false-negative or unexpectedly reduced activations within task-activated networks may result in inadvertent surgical resection of the critical network cortical regions, leading to postoperative disability, or may in some cases lead to overly cautious conservative resection, if necessary complementary intraoperative mapping cannot be performed in these patients [7,42]. Additionally, the presence of NVU may adversely affect hemispheric language dominance determination, since false negative ipsilesional activation can lead to erroneous assumptions regarding contralateral dominance or even apparent but erroneous cortical reorganization, as first suggested by Ulmer and colleagues [43]. Since the potential for actual cortical reorganization, including contralesional reorganization, is certainly present within the language network as well as in other brain networks, NVU detection is very important to differentiate true cortical reorganization from apparent but actually “pseudo”-reorganization in the setting of brain tumors [44,45]. For all of these reasons, NVU detection and eventual mitigation are clinically important. The potential for NVU should therefore be taken into account during interpretation of presurgical fMRI activation maps in order to avoid potential postsurgical complications. Figure 1 shows sensorimotor mapping using tb-fMRI with demonstration of neurovascular uncoupling in a patient with a left perirolandic glioblastoma. In this case, complete absence of left primary motor cortical activation necessitated the performance of intraoperative cortical stimulation mapping for adequate detection of the viable electrically active (functional) motor cortex.

## 7. NVU Assessment on Resting-State fMRI

In recent review articles, Agarwal and colleagues provided a detailed review of studies performed on various rs-fMRI metrics which were investigated for assessment of NVU potential [40]. Retrospective rs-fMRI investigations on the sensorimotor network that were undertaken by our team have shown prevalence of NVU impacting this network in patients with brain tumors [46,47,48,49]. This previous review described procedures for collecting, analyzing, and monitoring rs-fMRI images as well as the effect of NVU on presurgical mapping including the language network. In previous investigations by our group, we assessed patients who received thorough clinical fMRI scanning for presurgical mapping at standard 3T and had de novo perirolandic low-grade gliomas [49]. Decreased BOLD signals within the ipsilesional sensorimotor cortex were observed in these patients on rs-fMRI analysis using seed-based as well as independent component analysis. This was a manifestation of NVU on rs-fMRI, since these patients were adequately performing the bilateral finger tapping task and had normal BOLD signal manifestation on the contralateral sensorimotor cortex; this was the first published report of NVU affecting rs-fMRI in low grade gliomas. On 7T rs-fMRI scans performed on two patients with de novo brain tumors, similar decrease in ipsilesional BOLD signals were noticed despite the greater signal-to-noise ratio of 7T MRI, which was insufficient to compensate for the NVU effect [47]. Furthermore, we investigated rs-fMRI ReHo maps and found a similar decrease of signals ipsilesionally owing to NVU induced effect on low frequency BOLD signals. In our research, we have demonstrated that the rs-fMRI ALFF could be a sensitive indicator for detecting NVU in brain tumors impacting the sensorimotor network [46]. Since rs-fMRI ALFF is regarded as a measure of neuronal activity [25,50], we used it to methodologically mitigate the effects of NVU on tb-fMRI activation maps of brain tumor patients [48].

Other groups have similarly investigated the effects of NVU on resting-state functional connectivity; one such study by Mallela and colleagues [51] explored alterations of functional connectivity in the hand motor cortex in both high and low grade gliomas. This study found greater decreases in functional connectivity in high grade gliomas due to NVU than in low grade gliomas.

Thus, it is important to stress that rs-fMRI and clinical tb-fMRI are both restricted by brain tumor-induced NVU [46,47,48,49]. It is necessary to further investigate brain tumor-induced NVU in rs-fMRI in order to implement rs-fMRI as a clinical presurgical mapping tool.

## 8. Cerebrovascular Reactivity Mapping (CVR) for NVU Detection in Brain Tumors

Although the precise pathophysiologic mechanisms responsible for NVU tumor-induced NVU are not yet fully understood, the potential for NVU can be identified by observing regional impairments in cortical cerebrovascular reactivity (CVR) [52]. CVR is the term used to describe the ability of cerebral blood vessels to either expand or contract in response to vasoactive stimuli, resulting in an increase or reduction, respectively, in regional cortical cerebral blood flow [53]. While the autoregulatory mechanism may remain intact in the tumor-affected vessels, it is functioning at its maximum threshold, i.e., the vascular bed is already maximally dilated to maintain adequate blood flow to the brain tissue. Consequently, the further augmentation of blood flow becomes depleted in the affected vessels. Perfusion or BOLD imaging techniques can be employed to measure CVR during challenges that elicit a vascular response [54,55].

BOLD signals are typically expressed as a percent change from the baseline (i.e., in the hypercapnia condition relative to normocapnia) because they are not readily convertible into absolute metrics such as blood flow per minute. The linear dependence of BOLD on cerebral blood flow (CBF) under respiratory challenges (i.e., hypercapnic conditions), makes it an effective way to measure CVR [56]. Hypercapnia can be induced by breath-holding (BH) or through inhaling CO_2_-enriched air. Previous studies have established robust correlations between CVR measurements obtained through BH and CO_2_ inhalation techniques [57]. It has been suggested that the BH method can serve as a straightforward and practical approach suitable for clinical implementation [52].

In a recent comprehensive study, Sleight et al. conducted a systematic review of 235 publications that investigated CVR methods using MRI [58]. Their findings indicated that CVR values obtained through MRI were comparable to or exhibited strong correlations with those derived from other imaging modalities such as positron emission tomography (PET). Furthermore, several studies examined and compared CVR measurements using different MRI techniques. Specifically, CVR measured through BOLD imaging and arterial spin labeling (ASL) exhibited a significant correlation when employing CO_2_ inhalation techniques [59,60,61,62].

### 8.1. CVR Mapping Using Breath-Hold Methods

Breath-hold (BH) BOLD fMRI is becoming a popular method for mapping CVR [63,64,65,66,67,68]. There are two types of BH tasks that have been widely adopted at major academic institutions in the United States, and each has its relative advantages and disadvantages. Both approaches utilize block designs with alternating blocks of breath-holding and normal breathing, and both approaches utilize the general linear model for fMRI analysis, contrasting a hypercapnia condition with a normocapnia condition. Both methods will be described below in detail.

The particular BH CVR paradigm developed by Pillai and colleagues at Johns Hopkins includes normal breathing periods of 40 s followed by a 4 s block of inspiration that immediately precedes a 16 s breath-hold period [63]. This cycle is repeated four times, and at the end of the last breath-hold period, an additional normal breathing period of 20 s is added. Thus, overall task duration is approximately 4 min and 20 s. The 16 s breath-hold blocks are well tolerated by most patients, including even patients with substantial neurologic impairment. One advantage of this version of the breath-hold task compared to the alternate version described below is that the breath-hold blocks are shorter, and thus they allow for higher patient compliance in cases where patients may have difficulty with longer breath-hold blocks. This method is based on the respiratory response function as described originally by Birn and Bandettini in 2008 [69]. This provides a consistent baseline condition from which the BH signal response is extracted. BH CVR mapping is effective for evaluation of both high-grade and low-grade gliomas [63,64,65,66,67,68]. Respiratory bellows are used to ensure subject compliance for the breath-hold task [68]. The end-inspiration BH response function has an initial BOLD signal dip followed by a subsequent signal increase. Therefore, a modified hemodynamic response function must be used for this particular breath-hold paradigm based on the previously described respiration response function proposed by Birn et al., for CVR mapping as noted above [69]. The need for such modification of the HRF from the standard canonical HRF that is used for typical fMRI analysis is one limitation of this particular breath-hold paradigm. Figure 2 shows this particular breath-hold task consisting of four cycles with each cycle of 4 s of initial inhalation (red), 16 s of post-inspiratory breath-hold (orange), and 40 s of normal breathing (green).

For an illustration of how BH CVR maps might be helpful in NVU detection, please see Figure 1, which depicts severe NVU caused by a glioblastoma in a patient with intact right hand and tongue motor function but completely absent BOLD activation in the hand and face representation areas of the left primary motor cortex. The areas of absent left motor cortical activation correspond to an extensive area of regional decrease in CVR, in contrast to the contralateral cortex that displays both intact motor cortical activation and normal right hemispheric cortical CVR.

Other groups have published similar results using similar duration breath-hold fMRI tasks for measurement of vascular reactivity. For example, Iranmahboob and colleagues [70] have described a slightly different approach to measurement of vascular reactivity by considering “peak to trough” measurements from such tasks as a measure of CVR. This group used similar 16 s breath-hold blocks following 4 s of controlled inhalation, followed by 40 s blocks of normal breathing, identical to the paradigm described above. The authors specifically mention in their work that the contrast mechanism responsible for the CVR maps is distinct from the contrast mechanism of gadolinium enhancement. This is of importance in all types of gliomas and not just in high grade tumors.

Tumor vascularity has been extensively evaluated using T2* dynamic susceptibility contrast MRI perfusion imaging [64]. In high grade gliomas, tumor angiogenesis is associated with higher leakage-corrected relative cerebral blood volume and relative cerebral blood flow [35,71]. However, in the case of low-grade gliomas, hyperperfusion is typically not observed. Therefore, T2* DSC perfusion imaging may not be a dependable method for assessing NVU in low-grade gliomas [64]. Research has shown that BH CVR mapping is helpful in evaluating both high-grade and low-grade gliomas [63,64,65,66,67,68,72]. In-depth information about the BH CVR mapping in the clinical setting, including implications to presurgical mapping, was provided by Pillai and Mikulis [52].

An alternate, simplified task can also be employed for breath-hold fMRI exams, and has been well established at Mayo Clinic (Rochester, MN) [73]. This task is based on the premise that, for healthy adult volunteers, cortical BOLD signal resulting from a 20 s breath-hold will peak, on average, about 30 s after initiation of the breath-hold, and will return to baseline at about 40 s [69]. Consequently, a simple block paradigm can be created using 40 s epochs, each consisting of 20 s of free breathing followed by 20 s of breath-holding [73]. The task is presented visually with two text stimuli, one indicating “breathe normally” and the other stating “hold your breath.” Five and a half epochs are commonly employed with the task beginning and terminating with a 20 s free breathing block. Using this scheme, carbon dioxide-induced BOLD signal will peak during the middle of the free breathing blocks and will return to baseline just prior to the next breath-hold. As with all breath-hold fMRI tasks, it is important to remember during data analysis that the cortical BOLD response to a carbon dioxide stimulus is substantially delayed compared to the BOLD response from a motor or language task event. Based on Birn’s BOLD respiratory response function, this simplified breath-hold task can be easily processed using a variety of commercial or research fMRI software systems. In doing so, the free breathing periods are designated as the “active blocks” given that peak BOLD signal elevation is expected during those time periods. A notable exception is that the first 20 s free breathing block is discarded from analysis given that it was not preceded by a breath-hold. Consequently, no BOLD signal elevation is expected during that block.

This simplified breath-hold task has the advantage that it is straightforward to teach. Patients are merely instructed to relax and breathe normally during the free breathing block and to immediately hold their breath when prompted by the breath-hold visual stimulus, regardless of where they happen to be in their free breathing respiratory cycle. Moreover, the temporally compact block design allows for a greater number of breath-hold cycles over a given acquisition period with potential statistical benefit. Disadvantages include some patients’ inability to hold their breath for a full 20 s. In these instances, patients are instructed to hold their breath for as long as possible before resuming breathing. While this risks alteration of the expected BOLD respiratory response function, this small deviation from protocol does not usually compromise results. Another potential disadvantage of the simplified task is that, on some cycles, patients may not be prepared to hold their breath when the visual prompt to do so is presented. For example, this may occur when the breath-hold stimulus is presented immediately after a free breathing exhalation. This “stimulus surprise” factor may lead to a brief delay in the initiation of breath-holding for a given cycle. Finally, the task assumes return of BOLD signal to baseline within 40 s after the initiation of the breath-hold. While this assumption about the BOLD respiratory response function is true on average, there may be variation in the return to baseline for a given individual, potentially impacting derived statistical maps.

See Figure 3 for an example of clinical application of the Mayo version of the BH CVR task. The BH CVR map of this 52-year-old male patient who presented with a WHO grade 3, IDH-wildtype, MGMT-methylated anaplastic astrocytoma displays prominent NVU-related cortical and subcortical CVR reduction. Notice the regionally reduced CVR superior and lateral to the peripherally-enhancing centrally cystic/necrotic left frontal lobe mass, which resulted in reduced sensorimotor activation. Figure 3A depicts the BH CVR maps, while Figure 3B displays a corresponding DTI color fractional anisotropy map and anatomic postcontrast T1-weighted image.

Pinto and colleagues [54] recently published an article that offered systematic guidance for carrying out CVR mapping without using CO_2_ inhalation techniques. They examined CVR mapping methods based on MRI and CO_2_ fluctuations without gas challenges in this review, particularly the methodological facets of the breathing procedures and the accompanying data processing. They discovered that the body of published research strongly suggests that active breathing control, such as BH task-fMRI, can result in larger CBF variations and more reliable CVR measurements than naturally occurring breathing fluctuations as assessed in resting-state fMRI.

In a recent investigation, Cohen et al. [74] conducted a comparative study to assess the repeatability and sensitivity of BH activation and CVR mapping using two different MRI sequences: an advanced multiband multi-echo EPI sequence and an existing multiband single-echo sequence. Their study involved healthy volunteers. The findings revealed that the multiband multi-echo sequence significantly improved the strength and extent of BH activation, as well as the repeatability and reliability of relative CVR measurements, particularly in regions where the single-echo imaging sequence showed high signal dropout. These results suggest that employing multi-echo imaging can be a valuable approach for obtaining reliable CVR measurements.

Despite its advantages in exam simplicity and ease of use, BH fMRI has a number of drawbacks such as inter-subject variability of BH periods [75], motion artifacts [76], variable patterns of hemodynamic response to BH during end-inspiration versus end-expiration BHs [77,78], and slower hemodynamic response to BH stimulus compared to behavioral task-related BOLD signal changes. BH responses differ among populations depending on the depth of intake before the breath-hold [79]. Longer BH durations produce more robust BH responses [80,81], although in general there is a practical patient tolerance limit of 20 s duration of the BH period [81,82,83]. The peak-signal response occurs several seconds after the BH period, making the modeling of the BH response for CVR mapping a complex task [69,80,81,84,85,86]. Furthermore, acceptable BH fMRI repeatability, without the inclusion of end-tidal CO_2_ measures, may not be achievable [80].

### 8.2. CVR Mapping Using Exogenous Gas Delivery Approaches

CVR refers to the percentage alteration in BOLD signal per mm Hg change in end-tidal partial pressure of carbon dioxide (etCO_2_). During scanning, CO_2_ gas administration is carried out by inhalation of mixtures of CO_2_ and room air by individuals. The CVR may be calculated using the variation in the BOLD fMRI signal in response to variations in end-tidal relative pressures of carbon dioxide (etCO_2_) [87,88,89,90,91]. CVR is determined as ∆%BOLD/∆etCO_2_ by measuring signal intensity at resting and during a vascular challenge state. Using ASL-MRI [92] and the gold standard Diamox-challenged 15O(H_2_O)-PET [93], the link between CVR as a measure of cerebrovascular reserve capacity versus variations in cerebral blood flow has been verified.

A recent review article by Fisher [87] presents a comprehensive analysis comparing factors such as effectiveness and cost between the simpler respiratory circuits developed to administer predetermined concentrations of inhaled gases [90,94,95,96,97] and sophisticated computerized systems developed to incorporate enhanced dynamic etCO_2_ stimuli [91,98,99,100] and prospective targeting of pCO_2_ and/or pO_2_ levels [88,101]. In a technical review, Liu et al. [89] provide a detailed review of CVR MRI mapping techniques with CO_2_ challenges. It is important to note that CO_2_ inhalation techniques necessitate the use of MRI-compatible equipment, and the use of a breathing mask, which may not be available or may not be tolerated by all individuals, unlike self-paced BH CVR methods, leading to their exclusion from CVR measurements studies.

Fisher and Mikulis [102], in a recently published review paper, provided a full account of their two decades of work on creating a stimulus-response method for examining the physiology of vasculature. They concluded that even though cerebral vascular abnormalities can cause complicated patterns of blood flow distribution in the brain, a single repeating vasoactive stimulation can discriminate between “normal” and “abnormal” patterns of flow.

Champagne and Bhogal [103], in a recent work, provided information on the brain tissue-specific reaction to respiratory stressors at 7 T MRI. To better understand the variables influencing the variations in the temporal delays of the CVR response between tissues, the authors of this study combined respiratory challenges based on hypercapnic and hyperoxic conditions. Their conclusion highlighted that variations in the temporal delays of the CVR response could be influenced by intravascular CO_2_ gradients within the nearby blood vessels. These gradients, along with factors such as blood flow effects and membrane permeability, play a role in determining the diffusion rate within local tissues, subsequently driving the vasodilatory response.

### 8.3. CVR Mapping Using rs-fMRI

CO_2_-hypercapnic stimulation and breath-hold may not be feasible in all clinical situations. Rs-fMRI signals have been investigated as a potentially easier alternative CVR measurement method [104]. It has been suggested that a workable CVR biomarker that would be more efficient than BH techniques in a broader population is the resting-state fluctuation of amplitude (RSFA), a task-free measurement obtained from rs-fMRI scans [104,105]. While spontaneous neuronal activity and physiologic components are recognized to contribute to RSFA, variations in CO_2_ are not the ones primarily involved in BOLD signal fluctuations [106,107]. One potential solution to this problem involves use of the etCO_2_ time curve as a regressor to create the CVR map using the rs-fMRI data [108]. Additionally, the global rs-BOLD signal’s frequency spectrum was split out into a number of distinct frequency bands, and the temporal relationship between the signal timecourses within these bands and the etCO_2_ timecourse was then examined [107]. To create the resting-state-derived CVR map (rs-CVR), the correlation was greatest with the 0.02–0.04 Hz frequency band. A wider frequency range of 0–0.1164 Hz has been suggested for rs-CVR mapping in order to account for greater inter-subject variability, however [109]. Yeh and associates [110] investigated concordance between regression-based resting-state CVR and the BH CVR mapping in glioma patients. They showed that regression-based technique rs-CVR performed better than the RSFA.

Several investigations of BOLD amplitude variations during a CO_2_-hypercapnic challenge or BH found a substantial association between resting-state BOLD fluctuations and end-tidal CO_2_ fluctuations. The use of variability in respiration and its effect on BOLD fMRI as a CVR biomarker is a new discovery [107,111]. However, given that the outcomes of resting-state approaches are unpredictable, further research is necessary before they can completely replace BH procedures [112].

More recently, deep learning methods have been applied to assess cerebrovascular reactivity based on resting-state BOLD data. This promises to be a useful method for CVR assessment that may have the additional advantage of improved reproducibility when compared to other CVR methods, and preliminary work suggests that this approach may be viable not only in normal subjects but also in patients with vascular disease such as Moyamoya disease, as well as in patients with brain tumors [113].

## 9. Other Approaches for NVU Assessment

In the case of non-lateralized brain networks such as the sensorimotor network, one can assess the symmetry of activation on task-based fMRI; if unexpectedly decreased or completely absent activation is seen in the ipsilesional portion of the network in the setting of primary gliomas, one can infer the presence of NVU. However, this approach based on visual inspection of individual task-activated networks fails in the setting of lateralized networks such as the language network, where an internal reference standard (contralateral homologous activation) is not present for comparison. Another pitfall of this approach is that its use depends heavily on the experience of the person interpreting the activation maps as well as general knowledge of functional anatomy pertaining to individual brain networks; thus, this is more operator-dependent than other objective methods such as BH CVR or ALFF-based methods, which are both less operator-dependent and allow whole-brain evaluation.

An additional rarely-used approach to the assessment of NVU is known as functional field mapping, which was initially published in research applications in the human visual system [114]. This method compares the results of visual field evaluation via visual perimetry testing to those of visual cortical activation via fMRI. Regions of the visual cortex displaying intact neural function, i.e., preservation of visual field on perimetry testing, but not displaying corresponding BOLD activation on fMRI vision mapping, represent areas of NVU affecting the visual network. While this is an effective technique for NVU assessment in the visual network, it has not been applied to other brain networks for NVU assessment.

## 10. Methodological Approaches for Mitigation of Brain Tumor Induced NVU

### 10.1. Using rs-fMRI

In an initial investigation [48], our research team aimed to improve the detectability of motor task activation within the ipsilesional sensorimotor cortex affected by tumor-induced NVU using the ALFF derived from rs-fMRI. Contrary to the other rs-fMRI functional connectivity metrics, ALFF is commonly regarded as a neuronal activity metric [25]. ALFF encompasses both a lower frequency band component associated with CVR and a higher frequency band component linked to neuronal activity [50]. This suggests that ALFF may offer a potential means to mitigate the effects of NVU on tb-fMRI activation maps. Our findings revealed that the periodicity of the average motor task activation time course in both the contralesional and ipsilesional hemispheres of tumor patients exhibited similarity with lower amplitude of BOLD signal in ipsilesional voxels, indicating the impact of NVU in the tumor region. Therefore, we used an ALFF-based correction strategy in this study using ratios of contralesional mean ALFF values within regions of interest to ipsilesional measured individual voxel ALFF values, in order to better identify task activations in the individual sensorimotor cortical voxels impacted by NVU. The signal timecourses of the corrected voxels following application of this NVU mitigation method were similar in periodicity to those of the contralateral non-NVU-affected voxels within the same sensorimotor network, indicating high functional specificity of the newly generated ipsilesional activation.

See Figure 4 for an example of how this ALFF-based NVU mitigation method can be applied to a BOLD sensorimotor activation map in a right-handed 58-year-old patient displaying NVU secondary to a left perirolandic WHO grade 4, IDH wildtype glioblastoma.

### 10.2. Using BH-fMRI

Additional approaches that have been investigated to lessen the effects of NVU are the following: (1) A coherence-based simulation incorporating BH CVR and tb-fMRI activation using identical block designs [115]. (2) A voxel-based BH CVR calibration approach [67,116,117].

### 10.3. Cerebrovascular Dynamics of Gliomas vs. Temporal Dynamics of BOLD Signal

Neovascularization, vascular remodeling, and abnormal metabolite production result in impairment of the local cerebral vascular system in brain gliomas [118,119]. It is thought that the abnormal neurotransmitters and metabolites produced by gliomas are circulated into the vascular network distant from the tumor area, resulting in extensive NVU and vascular remodeling [120,121]. It may also alter vasculature of the contralesional hemisphere [122]. We do not yet fully understand the scope and clinical significance of glioma-induced widespread vascular modulation.

According to recent research, bulk blood flow may be detected by looking at the temporal patterns of low-frequency fluctuations in BOLD fMRI [123]. A considerable portion of the low frequency variation in resting-state BOLD fMRI data may be efficiently modelled as a single low-frequency signal with different delay time across the brain [124]. When compared to the delays that would be anticipated if the signal were travelling through the brain with blood as it moved through the vasculature, the pattern of relative delay periods in various parts of the brain remains consistent. These low frequency systemic, non-neuronal signals from rs-fMRI were referred to as systemic low-frequency oscillations (sLFO) by Tong and colleagues [125]. They proved that time-shift analysis of rs-fMRI signals may capture the slow sLFO transmission along the cerebrovasculature [126].

BOLD asynchrony was studied by Gupta and colleagues [127] to differentiate between high-grade and low-grade gliomas. They discovered that high-grade gliomas have advanced and magnified hemodynamic fluctuations; as a result, time-shift analysis may enable characterization of gliomas’ hemodynamic properties and help discriminate between high- and low-grade gliomas. To examine cerebrovascular fluctuations and their relation to tumor aggressiveness, Cai and colleagues [128] performed a time-shift assessment on the rs-fMRI of 88 glioma patients. They discovered that the gliomas caused grade-specific cerebrovascular dysregulation throughout the whole brain, along with changed sLFO signal time-shift characteristics. Glioma-related asynchrony in vascular movements was examined by Petridies and colleagues [129] to help identify tumors from healthy brains. They discovered that BOLD asynchrony was inversely linked to neuronal density and closely linked to total cellularity, tumor density, and cellular proliferation.

Yet another approach to the problem of overcoming the adverse effects of NVU on detectability of functional MRI activation is the development of novel MRI pulse sequences that may be more resistant to the effects of NVU than conventional EPI BOLD sequences. Current work in this area remains active, including development of new arterial spin labeling (ASL) and other advanced perfusion sequences that may allow dynamic assessment of functional status without reliance on conventional BOLD sequences that are susceptible to the effects of NVU [personal communication with Dr. David A. Feinberg, Univ. of California, Berkeley, and related to currently funded research of senior author JJP and Dr. Feinberg]. It is possible that such novel sequences may be used in addition to or in lieu of standard BOLD sequences in presurgical mapping exams in the future to better inform neurosurgeons regarding network nodes/hubs that may not be readily visualized on standard mapping.

## 11. Conclusions

In conclusion, with active ongoing investigation regarding new methods for detection and mitigation of NVU and associated alterations in regional hemodynamics in the setting of brain glioma presurgical functional mapping, we expect enhanced reliability of clinical fMRI in this setting. Hopefully, this will result in better understanding of the pathophysiology of NVU induced by brain tumors and result in better methods to overcome the adverse effects of NVU in presurgical brain mapping.

## Figures and Tables

**Figure 1 cancers-15-04473-f001:**
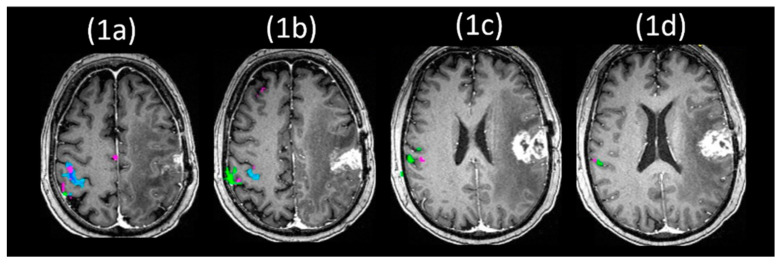
Sensorimotor mapping using task-fMRI with demonstration of neurovascular uncoupling. This 58-year-old male with an IDH-wildtype (MGMT status unknown) WHO Grade 4 glioblastoma (GBM) involving the left perirolandic region displays striking absence of activation in the left primary motor cortex and somatosensory cortex despite ability to move his hand and tongue sufficiently well to perform the motor tasks adequately. This corresponds to a large cerebrovascular reactivity (CVR) deficit comprising most of the left frontal lobe as shown on the overlaid breath-hold (BH) CVR map, and the discordance between actual motor function on neurological exam and visualized task-evoked activation is consistent with tumor-induced neurovascular uncoupling (NVU). Please note the well-observed activation of the hand and face motor areas in the right hemisphere but complete absence of activation in the homologous regions in the ipsilesional left cerebral hemisphere that is indicative of neurovascular uncoupling in this patient with intact bilateral motor function. Color-coding used for the composite motor activation map is the following: green represents the vertical tongue movement task; magenta represents the bilateral simultaneous sequential finger tapping task; blue refers to left hand opening/closing; and orange refers to right hand opening/closing. (1**a**–**d**) Depict anatomic sections from superior to inferior aspects of the pre- and post-central gyri. Anatomic (postcontrast sagittal T1-weighted and axial T2 FLAIR) images are depicted in (1**e**,**f**), respectively. (1**g**) Displays the BH CVR map in dark blue overlaid on postgadolinium T1-weighted anatomic images with superimposed task activation maps also displayed.

**Figure 2 cancers-15-04473-f002:**
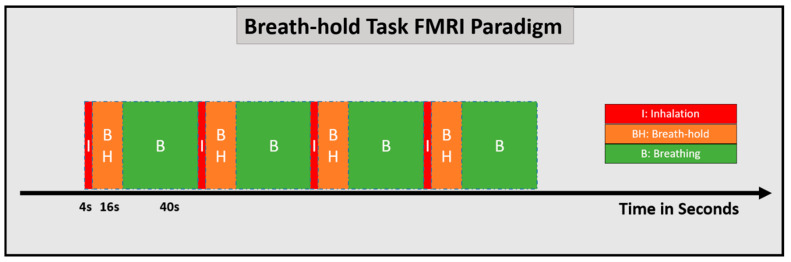
Breath-holding task fMRI design—four blocks, each consisting of 4 s of inhalation (red), 16 s of breath-hold (orange), and 40 s of normal breathing (green).

**Figure 3 cancers-15-04473-f003:**
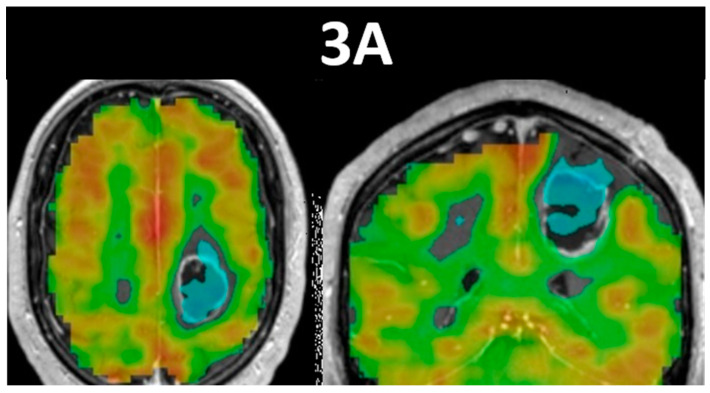
This figure displays breath-hold cerebrovascular reactivity (BH CVR) maps in the axial and coronal planes in (**3A**), and DTI color fractional anisotropy (FA) map and anatomic postcontrast T1-weighted image in the coronal and axial planes, respectively, in (**3B**). This is a 52-year-old male patient with Turcot Syndrome who presented with a WHO grade 3, IDH-wildtype, MGMT-methylated anaplastic astrocytoma that displays prominent NVU, as demonstrated by the Mayo simplified version of the BH CVR task. Notice the regionally reduced CVR superior and lateral to the peripherally-enhancing centrally cystic/necrotic left frontal lobe mass, which resulted in reduced sensorimotor activation. The spatial proximity of the mass to the corticospinal tract (in blue) and the cingulum bundle medially and superior longitudinal fasciculus laterally (in green) is shown in the coronal DTI color FA map. Notice the irregular peripheral enhancement and central cystic and/or necrotic nonenhancing region on the T1-weighted anatomic image.

**Figure 4 cancers-15-04473-f004:**
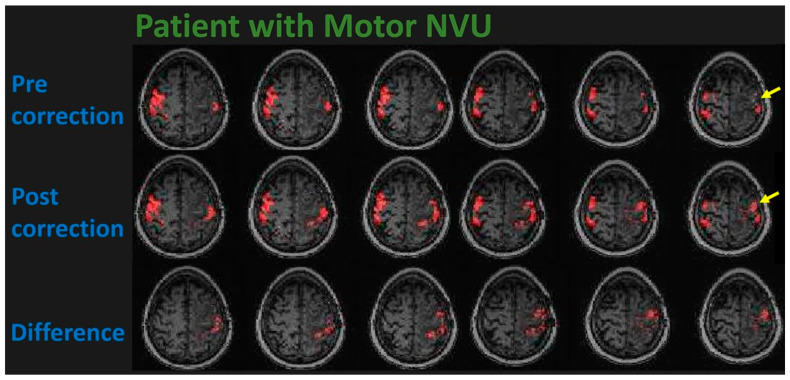
This figure displays fMRI activation maps in a 58-year-old strongly right-handed male patient with a left frontal lobe WHO grade 4, IDH-wildtype glioblastoma. Although the patient was able to move his right hand well enough to perform the finger tapping and hand opening/closing tasks, virtually no activation is seen in the left precentral gyrus corresponding to the hand representation area of the left primary motor cortex on the initial pre-correction activation maps shown on the (**top**) row (see yellow arrow pointing to area of expected but markedly reduced activation). The (**second**) row depicts the post-correction activation map that displays newly visible activation in the primary motor cortex after application of the resting-state amplitude of low frequency fluctuation (ALFF)-based NVU mitigation method. The (**third**/**bottom**) row depicts the difference between motor activation on the corrected and uncorrected maps, i.e., the newly-visualized activation resulting from the NVU mitigation method. Please note that the BOLD signal timecourses associated with the newly depicted voxels are identical to those of the originally visible ipsilateral and contralateral activated voxels, indicating high functional specificity of the new activation.

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
