# Peer review of "Detection and Mitigation of Neurovascular Uncoupling in Brain Gliomas"

_cancers, 2023, doi:10.3390/cancers15184473_

Round 1
Reviewer 1 Report
Thank you for allowing me to review “Detection and Mitigation of Neurovascular Uncoupling in Brain Gliomas” by Shruti Agarwal et al.
Summary: This is an outstanding review of an important topic by a leading group in the field. I only have some minor comments.
I suggest accepting this paper with minor revisions.
Specific comments:
I would expand the “3. Preoperative BOLD fMRI” section. A good review is PMID: 26848555. There are still those that resist pre-op fMRI, famously the French neurosurgeon H. Duffau. I would stress that pre-op fMRI can be considered the standard of care. There is a CPT code for billing and most insurance companies cover the procedure. Explain why language lateralization is important and why false negatives from NVU are important for this question: a false negative in left Broca’s Area can yield an incorrect language lateralization. For example, John Ulmer’s original paper PMID: 12591636.
Need to mention that cortical reorganization actually does happen (good reviews: PMID: 26848558; recent papers: 31333562,) and that one of the main challenges is to differentiate between “true” reorganization and “false” reorganization (or pseudo-reorganization, using Ulmer’s neologism) caused by NVU. This point will serve to make the outstanding work of the authors on overcoming NVU even more clinically relevant.
The first author may want to get some input from clinical colleagues on the above topics.
Page 4: “The possibility of false-negative or unexpectedly reduced activations within task-activated networks may result in inadvertent surgical resection of the critical network cortical regions, which can result in unexpectedly severe postsurgical neurologic disability (8, 47).” This is an overstatement. Neurosurgeons are rather conservative in resections, especially in terminal diseases such as gliomas. They almost never operate adjacent to eloquent areas without interoperative cortical mapping. Hence, the chance of inadvertent resection is rather remote.
Page 5: Section 7. NVU Assessment on Resting-State fMRI. Notwithstanding the leading work of the present authors in this field, it is a good idea to support this work though the inclusion of the work of other labs: PMID: 26250554 and 27457676.
“Since rs-fMRI ALFF is regarded as a measure of neuronal activity (30), we used it to methodologically mitigate the effects of NVU on tb-fMRI activation maps of brain tumor patients (54).” Is reference 54 correct here? It seems to be Bharat Biswal’s group not the Hopkins group.
Page 8, at the end of the first paragraph, the last sentence is incomplete. There may be a few sentences inadvertently deleted.
Page 13: “1) A coherence-based simulation incorporating BH CVR and tb-fMRI activation using identical block designs (71, 73)” Are these references correct for coherence mapping? The authors may have meant Voss et al 2019 PMID: 30991031
Instead of reference 9 (an abstract) suggest the published manuscript: 20708553 or 27383533
Reference 71. Zaca D, Jovicich J, Nadar SR, Voyvodic JT, Pillai JJ, editors. Cerebrovascular reactivity-based calibration of 725 presurgical motor activation maps to improve detectability of the BOLD signal in patients with perirolandic brain 726 tumors. International Society for Magnetic Resonance in Medicine (ISMRM) 21st Annual Meeting 2013 April 20-26, 727 2013; Salt Lake City, Utah. Are the authors “editors”? This seems like an abstract at the ISMRM.
Reviewer 2 Report
This article should be review to indicate NVU in glioma. However, the content are basic explanation of fMRI and CVR mapping. If author want to explain CVR magging, this article should be research article.
Round 2
Reviewer 2 Report
The manuscript is fine and acceptable to publish.